# Whistler echo trains triggered by energetic winter lightning

I. Kolmašová [1,2,4] ✉, O. Santolík [1,2,4] & J. Manninen [3]

Lightning generated electromagnetic impulses propagating in the magnetospheric plasma disperse into whistlers – several seconds long radio wave signals with decreasing frequency. Sometimes, multiple reflections form long echo trains containing many whistlers with increasing dispersion. On January 3, 2017, two necessary prerequisites – a pronounced lightning activity and a magnetospheric plasma duct – allowed for observations of a large number of whistler echo trains by the high-latitude station in Kannuslehto, Finland. Our investigation reveals that the duct existed for nearly eight hours. We show that causative lightning sferics arrived to the duct entry from three different winter thunderstorms: a small storm at the Norwegian coast, which produced energetic lightning capable to trigger echo trains in 50% of cases, and two large storms at unexpectedly distant locations in the Mediterranean region. Our results show that intense thunderstorms can repetitively feed electromagnetic energy into a magnetospheric duct and form whistler echo trains after subionospheric propagation over distances as large as 4000 km.

Long lasting trains of sounds heard from electromagnetic recordings at audible frequencies have been reported as series of swishes following each other with almost perfectly regular spacing of a few seconds, the train persisting on occasion for as long as a few minutes[1]. Groups of these whistling atmospherics were analysed to conclusively link their origin to lightning discharges[2] and to show that the electromagnetic pulse generated by rapidly varying lightning stroke current propagates through the plasma medium surrounding the Earth and returns back in the dispersed form of whistlers[2]. A rare observation of extremely long whistler echo trains with a low decrement of intensity was reported over a band about 1 kc/s wide centred on a frequency between 3 and 4 kc/s, indicating that some form of focusing action is taking place[2]. A case was also reported, in which many echo trains were repetitively heard for almost 4 h[3]. The corresponding focusing action was identified[4,5] with refractive effects of field-aligned ducts of enhanced plasma density, where whistlers bounce between the hemispheres, reflecting at each bounce above the ionosphere, with some of the energy leaking through to the ground[6,7]. Conditions for such propagation of whistlers, leading to observations of whistler echo trains, were found to be favourable on the low-latitude side of the knee

in the magnetospheric ionization[8], also known as the plasmapause. Short echo trains were subsequently observed by spacecraft in the magnetosphere[9–11]. Ray tracing simulations showed that both the radial profiles of density enhancement[12] with an equatorial width of 0.02–0.05 Earth radii and the density enhancement factor larger than 10% are needed to enable ducted propagation between the two hemispheres at middle latitudes[13]. The duct duration was estimated on the order of one day[14], or several hours[15,16]. Numerical studies of the propagation of very low frequency whistler waves in magnetospheric plasma ducts demonstrated that these ducts can be formed by the field-aligned density enhancements (high-density ducts) or density depletions (low-density ducts)[17].

The origin of the ducts is, after many years of research, still under debate. It can be related to, ionospheric E-region irregularities[18,19] or instabilities[20], or, possibly, to the thundercloud electric fields[12,21–25]. Recent modelling results[26] confirm that plasma ducts can be generated by atmospheric gravity waves, which perturb the zonal and meridional neutral winds, subsequently influencing plasma motion and related electric fields. As a result, corotating electron density irregularities in the plasmasphere can be produced. However, it cannot be excluded

[1]Department of Space Physics, Institute of Atmospheric Physics of the Czech Academy of Sciences, Prague, Czechia. [2]Faculty of Mathematics and Physics, Charles University, Prague, Czechia. [3]Sodankylä Geophysical Observatory, Sodankylä, Helsinki, Finland. [4]These authors contributed equally: I. Kolmašová, O. Santolík. ✉e-mail: iko@ufa.cas.cz

that quasi-electrostatic fields from thunderclouds could drive duct formation in some situations[12]. The most straightforward natural method of confirming the presence of ducts and studying their properties is through the observation of lightning whistler echo-trains. All of the whistlers within an individual echo train originate from one single lightning stroke. Each subsequent whistler in the train exhibits an increasing dispersion time as the distance travelled through the dispersive plasma medium grows with each bounce. Interestingly, properties of the causative lightning strokes, which generate radio waves serving as duct-probing tools, have never been thoroughly investigated.

It is believed that each ducted whistler may precipitate radiation belt electrons into the conjugate ionospheric region and lower atmosphere[27]. A notable synchronization between sferics and whistler echoes was observed in the past[28], leading to a suggestion that new lightning discharges can be triggered from whistler echoes. The follow-up study supported this hypothesis showing that electron precipitation caused by a group of lightning whistlers might have triggered an atmospheric discharge producing another group of whistlers[29]. Nevertheless, a mechanism describing a role of precipitating electrons in modifying electrical properties of the thundercloud and triggering lightning initiation has been never proposed.

In this work we present results of a detailed analysis of lightning strokes, which produced numerous whistler echo trains detected by the high latitude station Kannuslehto, Finland during nearly eight hours on January 3, 2017. The investigated source lightning strokes originated from two large winter thunderstorm systems occurring in the Mediterranean region and one small-scale storm hitting the southwestern Norwegian coast upon arrival of the cyclone Axel. The impulsive radio signals of sferics from lightning strokes in these different storms were able to enter the same duct and trigger the whistler echo trains even after four thousand kilometres long travel in the Earth-ionosphere waveguide. The electric fields from distant thunderclouds were most likely not involved in the duct formation and echo whistlers did not trigger additional lightning strokes. The strength of whistlers was dependent on the strength of corresponding sferics, which were weaker for more distant strokes. Unusually large portion (41%) of causative strokes were positive, nevertheless, the strength of whistlers was not dependent on the polarity of causative discharges. We demonstrate that winter thunderstorms with a high incidence of energetic lightning can supply enough energy to the magnetosphere and produce strong whistlers bouncing for a long time if a long lasting duct is formed.

## Results
### Whistler echo trains
After the midnight on January 3, 2017, the high latitude receiving station in Kannuslehto[30], Finland (67.7°N, 26.3°E) started to observe intense echo trains of lightning whistlers (See Methods, Subsection Receiving station). This was clear evidence of a formation of a plasma duct. Examples of such trains of whistlers identified in the frequency-time spectrograms are illustrated in Fig. 1. A white vertical line at the beginning of each train corresponds to an impulsive broadband signal of sferics emitted by the causative lightning discharge into the Earth-ionosphere waveguide.

A detailed look at the immediately following whistlers of the trains reveals broad and fuzzy traces, strongly suggesting a multipath propagation of signals within a single wide duct, or in many smaller ducts, resembling a bundle of spaghetti[31]. A very similar succession of whistlers in the three spectrograms suggests that all these echo trains propagated in the same duct or a system of ducts, probably located close to the receiving station. For a field-aligned duct it means that it would be close to the magnetic field line, which passes through Kannuslehto and intersects the geomagnetic equator at a radial distance of 5.5 Earth's radii in the dipole approximation, i.e., close to an L shell of 5.5.

An inspection of all Kannuslehto measurements[32–35] from January 3, 2017 revealed 151 whistler echo trains with their causative sferics. We also found 10 trains, which lacked sferics at their beginning. This indicates that the causative discharges might have occurred in the opposite hemisphere and the emitted signals entered the opposite end of the duct.

We examined the list of lightning detections provided by the European lightning detection network EUCLID (EUropean Cooperation for LIghtning Detection)[36,37] and the global lightning detection system WWLLN (World Wide Lighting Location Network)[38] to identify causative strokes and determine their locations. For all lightning candidates, we meticulously verified the timing, accounting for the propagation delay that the signals accumulated during their journey from the source lightning location to the receiving station. We excluded all echo trains for which we were unable to identify the location of their source lightning strokes. Additionally, we excluded the whistler trains with undetected sferics, as we did not find any corresponding lightning detections in the southern hemisphere within several thousands of kilometres from the location magnetically conjugated to Kannuslehto using the global WWLLN data.

Our final dataset consists of 135 trains recorded from 01:28:24 UT to 09:04:37. The number of recognizable echoes varied from one (the condition for including the event in our dataset) to about thirty. The lightning whistlers bounced between hemispheres more than ten times in one third of events. One fifth of trains was overlapped by another train and the number of echoes cannot be estimated. Eight echo trains split after a few bounces in two trains, which occupied different frequency bands. The interval between consecutive trains varied significantly from 5 s to 38 min with an average value of 3.4 min. If we took into consideration all 161 echo trains (including these without visible sferics and without identified causative strokes), an echo train appeared in the duct on average every 3.1 min.

We divided the echo trains in four groups according to the visually estimated maximum power spectral density (PSD) of the magnetic field of the first echo trace (extra strong: $PSD$ = about $10^{-4}$ pT$^2$/Hz, 17 trains; strong: $PSD$ = about $5 \cdot 10^{-5}$ pT$^2$/Hz, 14 trains; middle intense: $PSD$ = about $10^{-5}$ pT$^2$/Hz, 27 trains; weak: $PSD$ = about $5 \cdot 10^{-6}$ pT$^2$/Hz, 76 trains). The intensity of echo whistlers and the upper frequency of the whistler trace decreased toward the end of the trains. Identifying individual whistler traces became more challenging towards the train ends, as the whistlers typically merged into a hissy emission, centred at about 3–4 kHz.

### Lightning activity
To gain an overview of the lightning activity, and to contextualize the locations of lightning strokes responsible for whistler echo trains meteorologically, we examined the lists of EUCLID and WWLLN detections for lightning, which occurred on January 3, 2017 from midnight to 10 UTC, well before and after the appearance of the echo trains. EUCLID detected 2316 strokes in a range of longitudes from 0 to 40°E and latitudes from 30°N to 70°N, while WWLLN detected in the same area 4332 lightning discharges. The maps showing these detections are displayed in Supplementary Fig. S1a, b colour-coded by their time of occurrence. Note that EUCLID does not cover the Eastern and African Mediterranean region, and that WWLLN detected fewer strokes in the area covered by EUCLID. (See Methods, Subsection Lightning location networks detections).

To create a consolidated list of individual cloud-to-ground (CG) lightning strokes, we excluded 1280 intracloud detections from the EUCLID list, considering that WWLLN is primarily sensitive to CG lightning. We identified 654 double detections as those lightning events detected by both networks within 1 μs, and less than 1° in latitude and longitude apart from each other. From these double detections, we retained only the EUCLID detections in the consolidated combined list, which now comprises 4714 lightning strokes, as shown

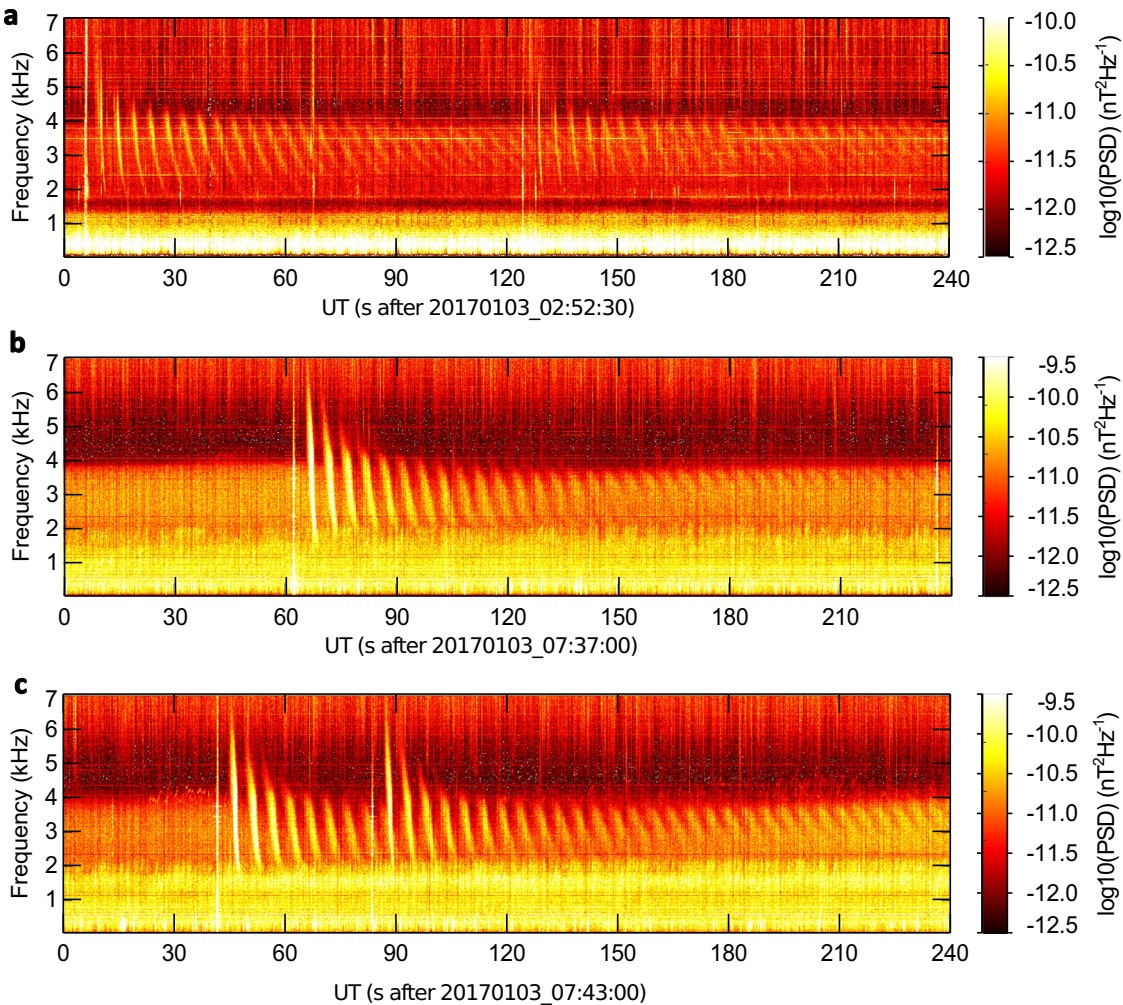

**Fig. 1 | Frequency-time spectrograms with examples of whistler echo trains recorded on January 3, 2017.** The colour represents the logarithm of the Power Spectral Density (PSD) in nT²Hz⁻¹. The white vertical lines, spanning over the entire frequency band and initiating each group of dispersed signals, are attributed to sferics emitted by lightning return strokes, which triggered the echo trains. Weaker broadband impulsive signals extending from the top of the spectrograms correspond to sferics emitted by distant lightning. Two overlapping whistler echo trains can be recognized in panels **a** and **c**. It is noteworthy that the intensity of VLF noise in the background is diminished within the whistler trains in panels **b** and **c**. This intriguing phenomenon was explained by a modification of electron distribution in the noise generation region caused by the intense whistler trains[55].

in the map in Fig. 2a, colour-coded by their time of occurrence. In panel b, the locations of source lightning strokes for echo trains are shown in colour, coded by the time of occurrence, while the locations of other lightning discharges are represented by smaller grey dots. Three separated thunderstorm systems can be identified and enclosed by coloured ovals in Fig. 2b. The largest area, indicated by the orange oval, was dominated by an extensive storm system over the Eastern and African Mediterranean Sea. The green oval represents two thunderstorms in Central Mediterranean moving southeast along the Apennine Peninsula and the western coast of the Adriatic Sea. A third thunderstorm struck the Norwegian coast in the morning hours as a consequence of the arrival of Cyclone Axel. The sferics travelled to the Kannuslehto receiving station about one thousand kilometres from the Norwegian western coast, up to three thousand kilometres from Central Mediterranean and up to four thousand kilometres from the Eastern and African Mediterranean region.

**Lightning sferics**

Figure 3 shows 8-ms long details of several sferics waveforms recorded at Kannuslehto. In all panels, the north-south component of the magnetic field is represented by the black line and the west-east

component is plotted by the same coloured lines which we used for distinguishing the thunderstorm systems. We are able to recognize the ground wave pulses travelling from lightning return strokes straight along the Earth surface and arriving first, followed later by the sky wave pulses propagating through one or more bouncing reflections from the bottom of the ionosphere and the Earth surface and, finally, by oscillatory signatures[39] – tweeks – resulting from a continuous superposition of many such bounces and represented by propagation modes of the Earth-ionosphere waveguide. Surprisingly we also noted the signatures of lightning initiation in a form of so called preliminary or initial breakdown pulses[40] preceding the ground wave.

Knowing the angle of arrival of individual sferics to the receiving station, the polarities of the ground wave in the two components also reflect the direction of the source lightning current and hence, the polarity of the source lightning. The sferics in panels a) and d) were emitted by positive lightning strokes transferring positive charge from the upper part of the thundercloud to the ground, the waveforms in panels b) and c) belong to negative lightning transporting negative charge from the cloud to the ground. The time delay between the sequences of preliminary breakdown pulses and the ground wave is shorter in case of negative lightning in panels b) and c), which is

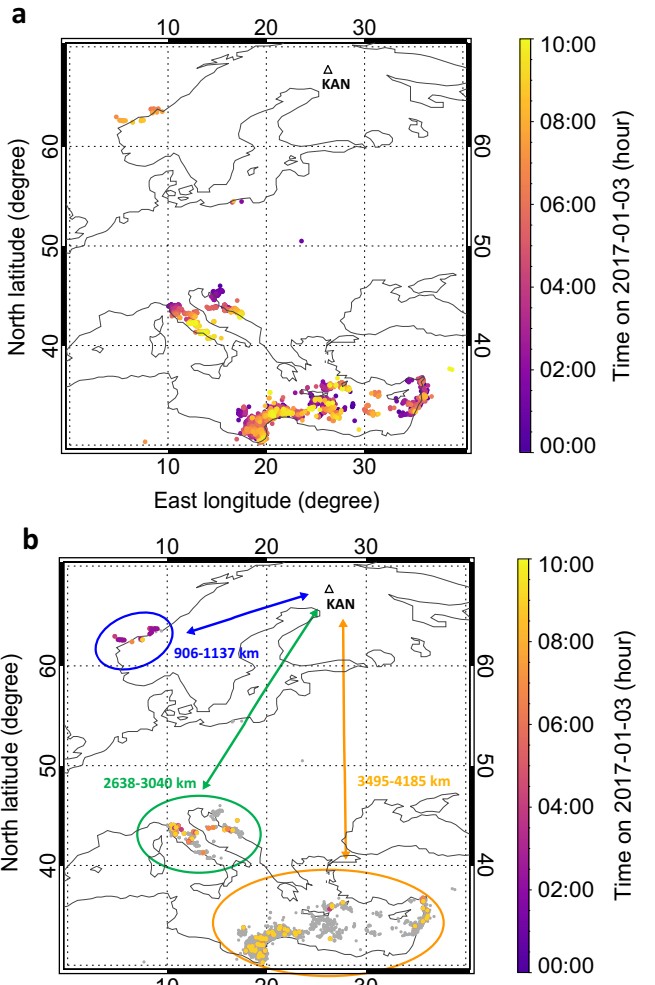

**Fig. 2 | Lightning maps from January 3, 2017. a** Locations of lightning discharges from the consolidated list consisting of EUCLID and WWLLN detections. **b** The locations of lightning discharges from the consolidated list (in grey) are shown, with the source lightning discharges for echo trains in colour coded by the time of their occurrence. The colours correspond to the time of lightning occurrence. The coloured ovals mark three separated areas with the lightning activity (Northern thunderstorm – blue, Central Mediterranean – green, and Eastern and African Mediterranean – orange). The intervals of distances the sferics travelled to the receiving station Kannuslehto (KAN) are also shown (906–1137 km from the Northern thunderstorm; 2638–3040 km from the Central Mediterranean; 3495–4185 km from the Eastern and African Mediterranean). Source data are provided as a Source Data file. The maps were plotted using the IDL® procedures (a product of Exelis Visual Information Solutions, Inc., a subsidiary of Harris Corporation).

expected based on different previous measurements conducted close to the lightning[40]. In case of very strong lightning the whole lightning process from the initiation to the return stroke is extremely short and the sequence of initiation pulses appears just before the return stroke pulse[41].

The sky waves of the west-east component of the magnetic field in panels 3a, b saturated the receiver as indicated by horizontal dashed grey lines. In case of a closer source lightning (Fig. 3d), both the ground wave and sky waves of the north-south component of the magnetic field saturated the receiver. To define the ground wave amplitude, the antenna pattern was considered, and the stronger component was divided by the cosine of the angle between the azimuth of arrival of individual sferics and the corresponding antenna plane. If one of the

components was saturated then the weaker non-saturated component was used in this procedure. In the case of the six strongest lightning strokes from the Northern storm, both components were saturated by the ground wave, and the ground wave amplitude was thus under-estimated. This received amplitude of the ground wave is related to the strength of the first whistler echo, as it is illustrated in Fig. 4a. We see that strong whistler echoes are received only for the strongest sferics, and that weak sferics produce weak echoes. This observation is consistent with the assumption that the duct entry is close to the receiving site.

We used the polarities of all the ground waves in the two recorded components to determine the polarities of the source discharges. From the total number of 135 sferics triggering the whistler echo trains, 56 belonged to positive CG discharges, 79 to negative ones. Note that these polarities are, without any exception, consistent with the results of the EUCLID detection network, wherever its data are available. The distribution of source lightning polarities in the three thunderstorm regions is illustrated in Fig. 4b, where the polarity is represented by dark pink full circles (−) and light pink crosses (+). The amplitude of the ground wave is here normalized to 1000 km, assuming that the amplitude is inverse proportional to the distance. We therefore neglect the additional attenuation caused by the finite conductivity of the Earth surface, which is dependent on the travelled distance but also on the strongly varying ground conductivity along the propagation path[42]. We also neglect effects of the finite conductivity of the lower reflective layer of the ionosphere, its spatial profile and temporal variations, which can be directly affected by underlying thunderstorms, by precipitating energetic electrons, or by X-rays originating in the solar flares[43]. Our procedure necessarily results in the underestimation of the normalized ground wave amplitudes of sferics travelling from more distant sources, as it is also suggested by generally weaker normalized amplitudes of sferics from more distant thunderstorms.

In order to contextualize the appearance of echo trains with lightning activity, we present the time sequence of ground wave amplitudes of sferics related to the observed echo trains in Fig. 5a. In Fig. 5b, we display lightning rates separately for the three thunderstorm systems as a function of time. The ground wave amplitudes are normalized to 1000 km in panel 5a. The arrows show the approximate time when the sun came up at locations of individual storm systems.

For comparison, we analysed forty-five sferics produced by energetic lightning (taken from the EUCLID list of discharges with peak currents exceeding 100 kA) without corresponding whistler trains. Surprisingly, to this group belong also the sferics emitted by the two strongest discharges from EUCLID (with a peak current of +316 kA) and WWLLN (a superbolt with an energy of 1.6 MJ) datasets. Their waveforms are displayed in Fig. 6.

## Discussion

The lightning activity was unusually intense and widespread considering the middle of the winter season in Europe[44]. Normally, winter months account only for 3% of the annual lightning in Europe[37]. Indeed, temperatures colder than the climatological mean prevailed in most of southern and southeastern Europe, while Ireland, northern parts of the UK, and Northern Scandinavia experienced much warmer conditions in January 2017. These temperature anomalies, arising from atmospheric circulation anomalies over the northeast Atlantic due to the warming climate[45], played a crucial role in creating favourable conditions for thundercloud formation in both Southern and Northern Europe. This climatological anomaly provided the opportunity to investigate in detail properties of lightning and their sferics capable of triggering whistler echo trains.

Three different thunderstorm systems overlapping in time but separated in space produced enough energetic lightning supplying the duct by lightning sferics, which bounced in the plasmasphere along the

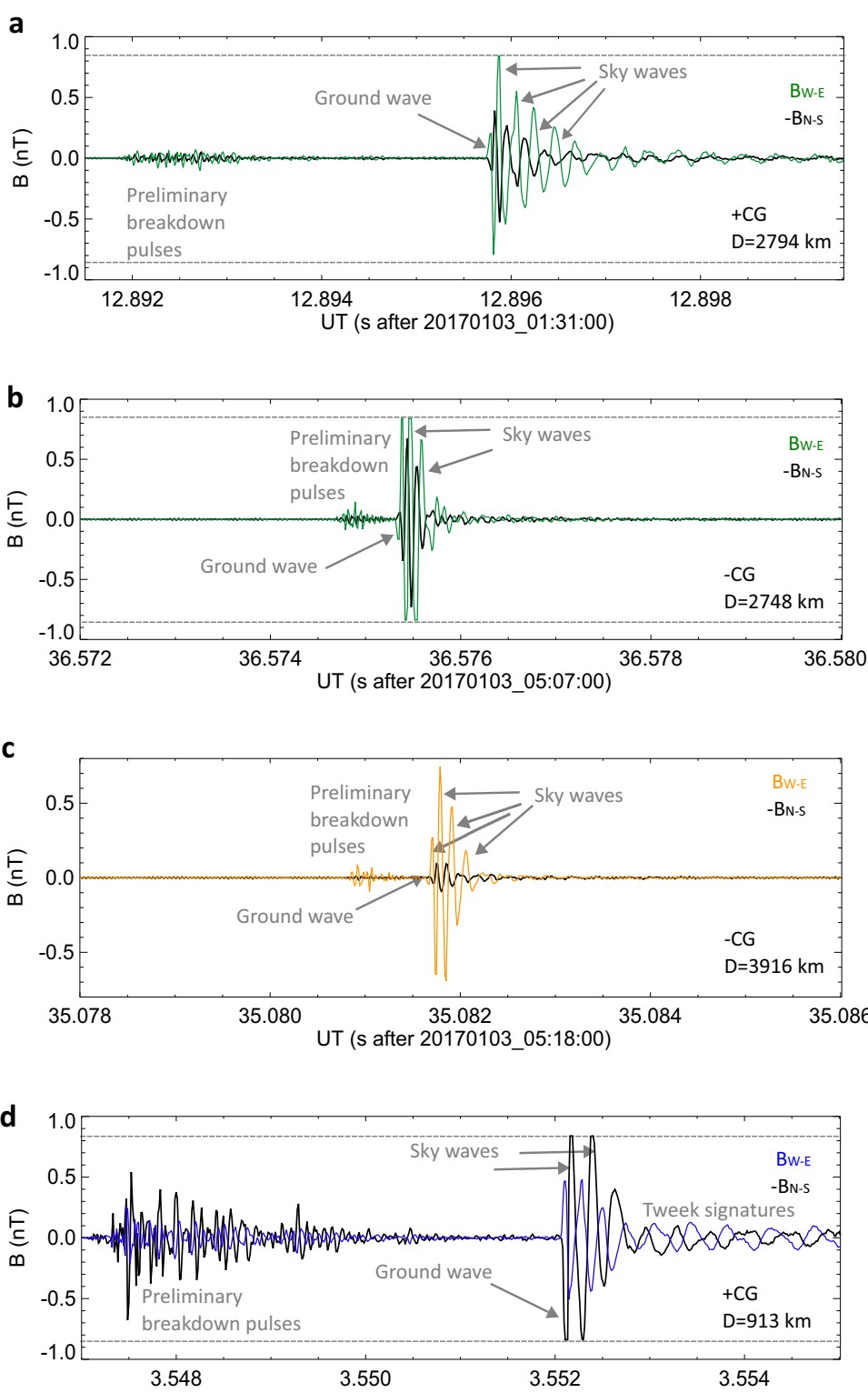

**Fig. 3 | Examples of lightning sferics. a** Waveform of the lightning sferics radiated by a positive CG occurring on 01:31:12.886259UT at 10.3951°E and 44.0589°N. **b** Waveform belonging to the lightning sferics radiated by a negative CG detected on 05:07:36. 565981 UT at 14.6317°E and 43.8039°N. **c** Waveform of the lightning sferics emitted by a negative CG occurring on 05:18:35. 068414 UT at 19.9121°E and 32.6886 °N. **d** Waveform of the lightning sferics radiated by a positive CG occurring on 06:42:03. 548862UT at 8.7698°E and 63.7329°N. The N-S component of the magnetic field is shown by the black line and the W-E component is plotted by coloured lines (Northern thunderstorm – blue, Central Mediterranean – green, and Eastern and African Mediterranean – orange). The N-S component is inverted in all panels to prevent the overlapping of curves. Grey dashed horizontal lines indicate the saturation level of the receiver. The ground waves, sky waves, preliminary breakdown pulse trains and tweek oscillations are identified in individual panels. Source data are provided as a Source Data File.

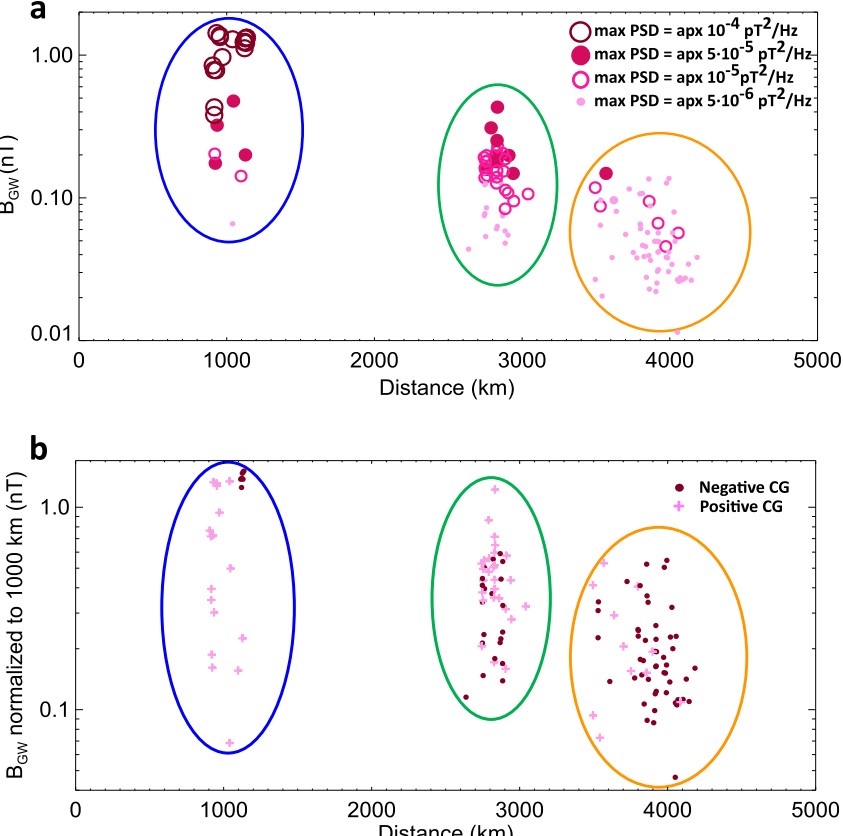

**Fig. 4 | Properties of sferics ground wave amplitudes. a** Intensity of the magnetic field of the sferics ground waves measured at the receiving station as a function of the distance from the causative strokes. The different symbols in shades of pink-purple colour correspond to the strength of the first echo whistlers (extra strong: max Power Spectral Density (PSD) = apx $10^{-4}$ pT$^2$/Hz, large dark purple open circles; strong: max PSD = apx $5 \cdot 10^{-5}$ pT$^2$/Hz, middle size purple full circles; middle intense: max PSD = apx $10^{-5}$ pT$^2$/Hz, middle size dark pink open circles; weak: max PSD = apx $5 \cdot 10^{-6}$ pT$^2$/Hz, small light pink full circles). **b** Intensity of the magnetic field of the sferics ground waves normalized to account for an ideal propagation over a distance of 1000 km plotted as a function of the distance from the causative lightning stroke. The polarity of causative stroke is represented by the different symbols (dark purple full circles and light pink crosses respectively for negative and positive causative cloud-to-ground (CG) strokes). Coloured ovals denote the storm locations (Northern thunderstorm – blue, Central Mediterranean – green, and Eastern and African Mediterranean – orange). Source data are provided as a Source Data File.

magnetic field line in a form of whistler echo trains. Given the considerable distance of these thunderstorms from the receiving station (900–4200 km) and from the presumed duct location, we can dismiss the hypothesis that the electric fields of the thunderclouds are responsible for the duct formation[12,21–25] in our case. Similarly, it is also very unlikely that the echo whistlers are involved in the lighting initiation[28,29].

By comparing the characteristics of sferics propagating to the receiving station (Fig. 3) from different distances we were able to investigate their effects on the properties of echo trains. According to our knowledge, such analysis has never been done. To characterize the lightning discharges based on the properties of their sferics, we utilized the amplitudes and polarities of the sferics ground waves measured by both orthogonal magnetic loops in conjunction with information about the discharge location provided by the lightning location networks.

We discovered that the proportion of positive lightning responsible for whistler trains (Fig. 4a) is twice as large as their average fraction in European winter storms, which typically reaches about 20% from December to February[37]. We can explain this effect tentatively by a not fully understood fact that positive strokes are generally observed to be much stronger compared with negative strokes in winter[46]. Surprisingly, the lightning strokes producing the most energetic echoes during the storm at the Norwegian coast (Fig. 4a) were negative with peak currents of −289 kA, −260 kA, −229 kA and −183 kA.

We showed (Fig. 4b) that the PSD of the first whistler echo is clearly related to the intensity of corresponding sferics ground wave and thus the closest lightning produced the strongest whistlers. Among the more distant lightning were energetic discharges as well, but the sferics lost their strength during the propagation in the waveguide[47]. The locations of lightning discharges from the consolidated list (in grey) are shown in Supplementary Fig. S1c, with the source lightning discharges for echo trains highlighted by pink-purple colour coding based on the maximum power spectral density of the magnetic field of the first echo whistler.

The storm at the Norwegian coast exhibited very low average lightning rate of less than 1 lightning per 2 min, while flash rates generally average up to 2 min$^{-1}$ in non-severe storms, and with higher values, over 10 min$^{-1}$ in severe thunderstorms[48]. (Note that a flash might be by definition composed of several strokes). Notably, one half of northern lightning was capable to trigger echo trains because of the high strength of their sferics. The echo trains triggered by the northern lightning stopped to occur at 8:45, when the storm dissipated.

The lightning activity in the Mediterranean (Fig. 5, both parts represented by green and orange colours) started already before the first echo train appeared in the recordings at about 1:30 UT. It is not surprising that the largest storm system, which occurred in the Eastern and African Mediterranean (shown in orange) exhibited the highest lightning rate. The accumulation of echo trains from 4:30 to 6:00 clearly coincided with the peaks in the lightning activity in the whole

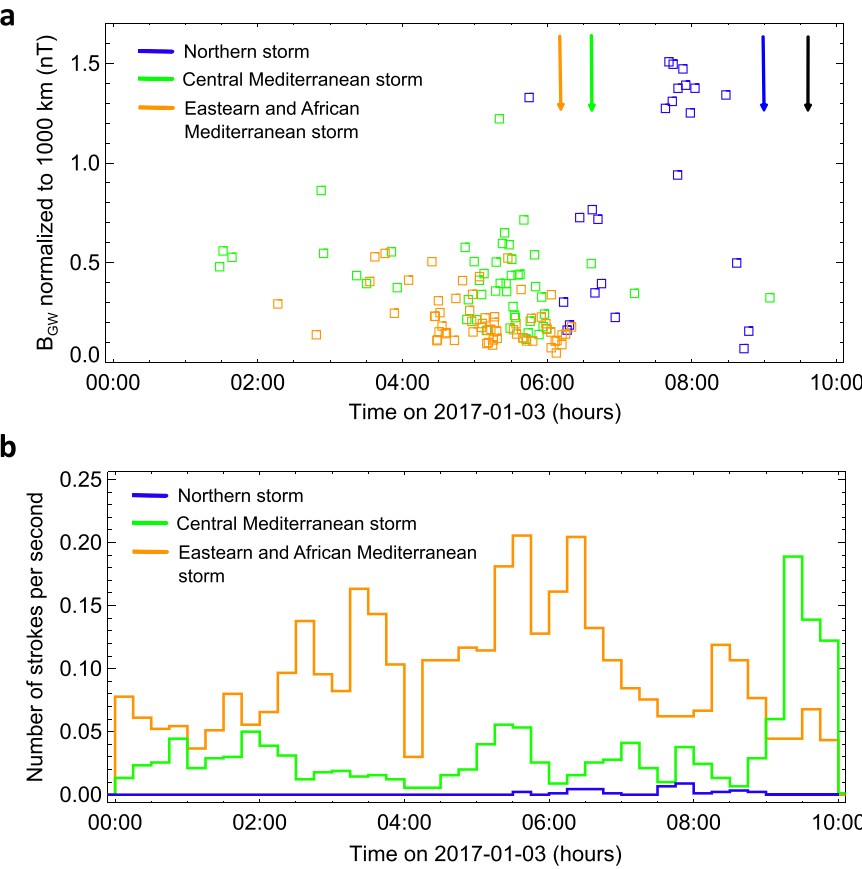

**Fig. 5 | Temporal overview of the lightning/whistler echo trains observations. a** Magnetic field amplitude of the sferics ground waves normalized to 1000 km as a function of time. **b** Lightning rates calculated separately for three thunderstorm systems. We use the same colours as in previous sections (orange for the Eastern and African Mediterranean storm system, green for Central Mediterranean, and blue for the Northern storm. The arrows indicate the time estimates of the sunrise at the storm system locations obtained using the NOAA tool (https://gml.noaa.gov/grad/solcalc/). The black arrow represents the time of the sunrise at the receiver location. Source data are provided as a Source Data file.

Mediterranean region. The trains triggered by the lightning from this largest storm stopped to occur around 6:10 UT due to the sunlight, which illuminated the propagation path of sferics and a denser dayside ionosphere[49] increased their attenuation. Nearly all echo trains related to the Central Mediterranean stopped arriving to the receiving station around 6:30UT, again with the sunrise at the storm location. The two trains observed after this time were triggered by very strong strokes and the attenuated sferics were still intense enough to launch the echo trains. No echo trains appeared after 09:36 when the Sun rose at the receiving station. This temporal analysis shown in Fig. 5 indicates that the duct has been living for at least seven and half hours, which is consistent with previous findings[14].

The analysis of sferics waveforms revealed additional noteworthy findings which merit attention: a) The initial sky waves were typically weaker than subsequent ones. This effect might have been related to the change of ionospheric reflection coefficients for different incident angles[50] of sky waves of different order, or due to waveguide mode conversion during the reflection[51]; b) The lightning initiation phase was identified in 71 % of waveform recordings. It was recognizable even in case of lightning occurring at a distance of 4000 km away from the receiver. Up to now, the longest documented distance from which the radiation produced by the initiation phase was received, was 2630 km[52]; c) Tweek oscillations were observed following only 21 sferics (15% of all events) and occurred exclusively during two periods (1:28 – 3:32UT; 6:13 – 7:48UT). One half of sferics produced by lightning from Norwegian coast were followed by tweek signatures, but only 5 sferics arrived with visible tweek oscillations from the Central

Mediterranean storm and 3 sferics from the most distant storm. Tweek signatures followed 2 negative CGs and 19 positive CGs. The lack of tweek signatures is quite surprising, as tweeks are known to be formed after about 1000 km of propagation in the night side waveguide[53] and were in the past observed to arrive also from distances larger than 5000 km[54].

This study clearly underscores the importance of investigating winter storms, especially those appearing in higher latitudes, which produce highly energetic lightning capable of generating strong whistler echo trains through which plasmaspheric ducts can be characterized.

## Methods
### Receiving station
The two-component VLF receiving station is operated by the Sodankylä Geophysical Observatory of the University of Oulu and is placed at a remote, low noise site. Electromagnetic VLF waves are captured in a frequency range from 0.2 to 39 kHz using two magnetic loop antennas with an effective surface area of 1000 m². These antennas are positioned orthogonally, aligning with the geographic north-south and west-east directions. The signals are sampled at a frequency of 78.125 kHz. The total voltage gain measured from the antenna termination to the input of the 24-bit AD converter is about 150 and the frequency response is flat above 1 kHz. The receivers are saturated at a signal strength of 0.85 nT. The raw magnetic field data, their format description and the relevant reading procedure are available at the data repository[33,32,34].

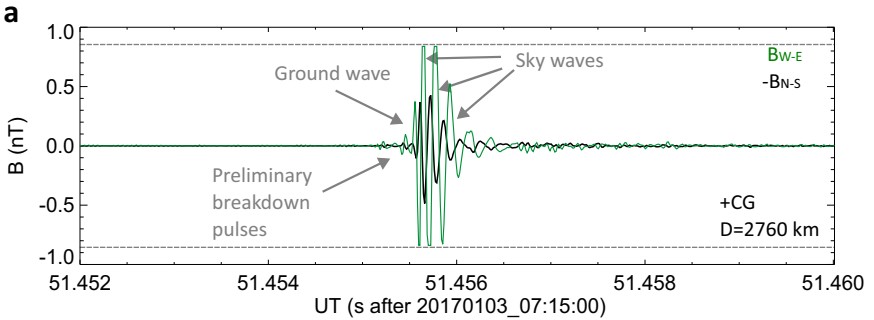

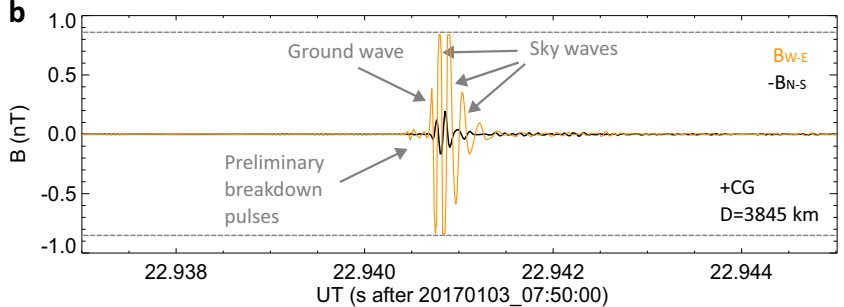

**Fig. 6 | Examples of sferics radiated by the strongest lightning strokes.**
**a** Waveform of the lightning sferics radiated by a positive CG occurring on 07:15:51.446161UT at 16.6281°E and 43.4542°N. **b** Waveform belonging to the lightning sferics emitted by a positive superbolt detected on 07:50:22. 927690UT at 21.4527°E and 33.2836°N. These sferics did not trigger whistler trains. The N-S component of the magnetic field is shown by the black line and the W-E component is plotted by coloured lines (Central Mediterranean – green, and Eastern and African Mediterranean – orange). The N-S component is inverted in both panels to prevent the overlapping of curves. Grey dashed horizontal lines indicate the saturation level of the receiver. The ground waves, sky waves, and preliminary breakdown pulse trains are identified in individual panels. Source data are provided as a Source Data File.

### Lightning location networks detections

The lightning location networks provide not only the time of occurrence and location of individual detected strokes but deliver also additional information about lightning polarities (EUCLID), lightning peak currents (EUCLID), types of lightning (EUCLID), or about radiated energies (WWLLN). The strongest stroke from the EUCLID list, with a peak current of +381 kA, hit Croatia close to Split. The most energetic stroke from the WWLLN list, a superbolt with an energy of 1.6 MJ, occurred near the coast of Libya. An empirical formula exists to convert the WWLLN energy into the corresponding peak current[38] for average lightning energies in the range of kilojoules and peak currents in the order of a few tens of kilo amperes. For higher WWLLN energies, the peak currents tend to be overestimated. For this reason, we refrain from combining these entities and focus on wave properties of sferics reflecting the strength and polarities of strokes responsible for the whistler echo trains. The list of lightning discharges included in our analysis together with their properties in provided in the Source data file.

### Data availability

The authors declare that all data supporting the findings of this study are available within the paper and its supplementary information files. Kannuslehto quick look plots are accessible from https://www.sgo.fi/Data/VLF/VLFData.php. Raw data used in this study in a form of one-hour continuous data files are available at https://data.mendeley.com/datasets/bhd4dt42mg/2 (part1) https://data.mendeley.com/datasets/gy27m8vy9v/2 (part2) https://data.mendeley.com/datasets/fvbp6j2fv8/1 (part3) at the Mendeley Data Repository. The World Wide Lightning Location Network (WWLLN) and EUCLID data analysed in this study are provided in the Source Data file. The WWLLN yearly data are available for sale from the University of Washington (http://wwlln.net/). For any information about obtaining the EUCLID lightning data please contact EUCLID Secretary: Dieter Poelman (dieter.poelman@meteo.be). (https://www.euclid.org/#). Source data are provided with this paper.

### Code availability

The custom code[35] for reading the raw data analysed in the manuscript was written in IDL® (a product of Exelis Visual Information Solutions, Inc., a subsidiary of Harris Corporation). It is available at https://data.mendeley.com/datasets/2kfk4vfgf7/1.

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

## Acknowledgements
We acknowledge the help of Dr. Gerhard Diendorfer with analysis of the EUCLID data. The work of I.K. and O.S. on the present paper has received funding from the GAČR grant No. 23-06430S. O.S. and I.K. also acknowledge funding from the European Union's Horizon Europe programme under grant agreement No. 101081772 — FARBES.

## Author contributions
I.K. and O.S. designed the study and analysed the data. O.S. wrote the data reading procedure. J. M. conducted the magnetic field measurements. The manuscript was written by I.K. and O.S.

## Competing interests
The authors declare no competing interests.
