## [Peer Review File · Nature Communications]

REVIEWER COMMENTS

Reviewer #1 (Remarks to the Author):

The paper describes analysis of several ELF whistler-mode signals produced by the lightning activity in the Mediterranean region and detected at the high-latitude station in Finland. The paper emphasizes the importance of understanding lightning occurring at high latitudes during winter storms, because they produce strong whistler echo trains, which can be used to understand magnetospheric density ducts.

This is an interesting paper and I do recommend it for the publication in Nature Communications with several minor corrections.

1. Lines 40-44: Quite extensive studies of the whistler-mode waves ducting by the density ducts in the magnetosphere by Streltsov et al. demonstrated that these ducts can be formed by the field-aligned density enhancements (high-density ducts) and density depletions (low-density ducts). So, please, include this information in the discussion in this part of the manuscript and add the references to:

Streltsov, A.V., M. Lampe, W. Manheimer, G. Joyce, and G. Ganguli (2006), Whistler propagation in inhomogeneous plasma, *J. Geophys. Res.*, 111, A03216, <https://doi.org/10.1029/2005JA011357>

2. Line 77: What is the magnetic L shell corresponding to the location of the Kannuslehto station? Is this location magnetically conjugate to the location of the storms identified from EUCLID or WWLLN measurements?

3. Lines 165-167: What is the physical difference between “ground” and “sky” signals shown in Figure 3? To my understanding, these are sferics which propagate in the Earth-ionosphere waveguide. What makes them different?

4. Line 174: What are “positive” and “negative” lightnings? Please specify the physical meaning of these terms, because not every reader of Nature Communications is familiar with this terminology.

5. Lines 202-203: You also do not include any effects of the ionospheric conductivity, which can also vary along the propagation path.

6. I think that the entire section “Methods” can be omitted from this paper. It contains general information which is not essential for the understanding of the main results of this paper.

Anatoly V. Streltsov

Reviewer #2 (Remarks to the Author):

The manuscript by Ivana Kolmasova and colleagues reports a detailed analysis of lightning discharges that produce whistler trains in a magnetospheric plasma duct. It is found that three thunderstorms in different locations produced lightning discharges >100 kA of negative and positive polarity that initiated the whistler trains. The largest surprises to me were that the wide ducts observed by the Van Allen probe satellites in space can possibly map down to lower ionospheric altitudes and that the first rise of magnetic field measurements can be used to determine lightning discharge polarities correctly, despite the ground wave normally being the most evanescent during long range propagation when compared to sky waves. It is therefore undoubtedly a remarkable study which deserves swift publication.

We thank A. Streltsov and the other reviewer for careful reading of the manuscript and for their helpful comments and suggestions. We responded to all of them and revised the manuscript accordingly. (The lines are related to the manuscript with tracked changes.)

Responses to Reviewers (in blue) with Reviewers' comments in black.

Reviewer #1 (Remarks to the Author):

The paper describes analysis of several ELF whistler-mode signals produced by the lightning activity in the Mediterranean region and detected at the high-latitude station in Finland. The paper emphasizes the importance of understanding lightning occurring at high latitudes during winter storms, because they produce strong whistler echo trains, which can be used to understand magnetospheric density ducts.

This is an interesting paper and I do recommend it for the publication in Nature Communications with several minor corrections.

1. Lines 40-44: Quite extensive studies of the whistler-mode waves ducting by the density ducts in the magnetosphere by Streltsov et al. demonstrated that these ducts can be formed by the field-aligned density enhancements (high-density ducts) and density depletions (low-density ducts). So, please, include this information in the discussion in this part of the manuscript and add the references to:

Streltsov, A.V., M. Lampe, W. Manheimer, G. Joyce, and G. Ganguli (2006), Whistler propagation in inhomogeneous plasma, *J. Geophys. Res.*, 111, A03216, <https://doi.org/10.29/2005JA011357>

We added the information about the types of density ducts on lines 46-48 and we added the recommended reference.

2. Line 77: What is the magnetic L shell corresponding to the location of the Kannuslehto station? Is this location magnetically conjugate to the location of the storms identified from EUCLID or WWLLN measurements?

The L-shell of Kannuslehto is 5.5. We added this information on line 91-94. We also added the information about the absence of lightning discharges occurring up to a distance of several thousands of kilometers from the point magnetically conjugated to Kannuslehto in the WWLLN lightning data on lines 106-108. EUCLID stations are distributed only across Europe and thus EUCLID covers only Europe.

3. Lines 165-167: What is the physical difference between "ground" and "sky" signals shown in Figure 3? To my understanding, these are spherics which propagate in the Earth-ionosphere waveguide. What makes them different?

We reworded the sentence to make clear that both the ground and thy sky wave propagate in the Earth – ionosphere waveguide, the first one straight along the Earth surface and the

second one through the reflections. The relevant sentence on lines 175-181 now reads as follows: "... We are able to recognize the ground wave **pulses** travelling from lightning return strokes straight along the Earth surface **and arriving first, followed later by** the sky wave **pulses** propagating through **one or more bouncing** reflections from the bottom of the ionosphere **and** the Earth surface and, **finally, by** oscillatory "tweek" signatures³³ resulting from **a continuous superposition of many such bounces and represented by** propagation modes **of the Earth-ionosphere waveguide...**"

4. Line 174: What are "positive" and "negative" lightnings? Please specify the physical meaning of these terms, because not every reader of Nature Communications is familiar with this terminology.

We added relevant information on the lines 187-189. The sentence now reads: "The sferics in panels a) and d) were emitted by positive lightning strokes transferring positive charge from the upper part of the thundercloud to the ground, the waveforms in panels b) and c) belong to negative lightning transporting negative charge from the cloud to the ground.

5. Lines 202-203: You also do not include any effects of the ionospheric conductivity, which can also vary along the propagation path.

We added following paragraph on lines 218-220: "We **also** neglect effects of the **finite conductivity of the lower reflective layer of the ionosphere, its spatial profile and temporal variations**, which can be directly affected by underlying thunderstorms, by precipitating energetic electrons, or by X-rays originating in the solar flares"

6. I think that the entire section "Methods" can be omitted from this paper. It contains general information which is not essential for the understanding of the main results of this paper.

Anatoly V. Streltsov

Based on the recommendation of the editor: "Keep current sections and provide further information about the analysis to ensure that the experiments can be reproduced without reference to other papers." we keep the Methods section and we also provide a list of analyzed lightning discharges and their properties in the Data Source file. The raw magnetic field data and relevant reading code will be available in a data repository.

Reviewer #2 (Remarks to the Author):

The manuscript by Ivana Kolmasova and colleagues reports a detailed analysis of lightning discharges that produce whistler trains in a magnetospheric plasma duct. It is found that three thunderstorms in different locations produced lightning discharges >100 kA of negative and positive polarity that initiated the whistler trains. The largest surprises to me were that the wide ducts observed by the Van Allen probe satellites in space can possibly map down to lower ionospheric altitudes and that the first rise of magnetic field measurements can be used to determine lightning discharge polarities correctly, despite the

ground wave normally being the most evanescent during long range propagation when compared to sky waves. It is therefore undoubtedly a remarkable study which deserves swift publication.

REVIEWERS' COMMENTS

Reviewer #1 (Remarks to the Author):

I am satisfied with the author's responses to my comments, and I do recommend this paper for the publication in Nature Communications in present form.